# Self-supporting sulfur cathodes enabled by two-dimensional carbon yolk-shell nanosheets for high-energy-density lithium-sulfur batteries

Fei Pei[1], Lele Lin[1], Daohui Ou[1], Zongmin Zheng[1], Shiguang Mo[2], Xiaoliang Fang[1] & Nanfeng Zheng [2]

How to exert the energy density advantage is a key link in the development of lithium–sulfur batteries. Therefore, the performance degradation of high-sulfur-loading cathodes becomes an urgent problem to be solved at present. In addition, the volumetric capacities of high-sulfur-loading cathodes are still at a low level compared with their areal capacities. Aiming at these issues, two-dimensional carbon yolk-shell nanosheet is developed herein to construct a novel self-supporting sulfur cathode. The cathode with high-sulfur loading of 5 mg cm$^{-2}$ and sulfur content of 73 wt% not only delivers an excellent rate performance and cycling stability, but also provides a favorable balance between the areal (5.7 mAh cm$^{-2}$) and volumetric (1330 mAh cm$^{-3}$) capacities. Remarkably, an areal capacity of 11.4 mAh cm$^{-2}$ can be further achieved by increasing the sulfur loading from 5 to 10 mg cm$^{-2}$. This work provides a promising direction for high-energy-density lithium–sulfur batteries.

[1] Pen-Tung Sah Institute of Micro-Nano Science and Technology, Xiamen University, Xiamen, Fujian 361005, China. [2] State Key Laboratory for Physical Chemistry of Solid Surfaces, Collaborative Innovation Center of Chemistry for Energy Materials, and Engineering Research Center for Nano-Preparation Technology of Fujian Province, College of Chemistry and Chemical Engineering, Xiamen University, Xiamen, Fujian 361005, China. Correspondence and requests for materials should be addressed to X.F. (email: x.l.fang@xmu.edu.cn) or to N.Z. (email: nfzheng@xmu.edu.cn)

Since the conventional Li-ion batteries (LIBs) based on intercalation compounds are already facing their energy density limit, researchers have been working to develop new energy-storage systems with high-energy density that can meet the ever-increasing requirement for portable electronics, electrical vehicles, and large-scale energy storage devices[1]. As the promising next-generation batteries, lithium–sulfur (Li–S) batteries with high-theoretical-specific energy ($\sim 2600$ Wh kg$^{-1}$) have been intensively pursued in the past few years due to the overwhelming advantages of the cathode material sulfur, such as high capacity (1675 mAh g$^{-1}$), low cost, widespread source, and nontoxicity[2–6]. However, the practical application of Li–S batteries is still limited by the low-sulfur utilization and poor cycle life, which mainly originated from the poor conductivities of sulfur and its discharge products $Li_2S$, the shuttling of the soluble intermediates ($Li_2S_n$, $4 \leq n \leq 8$) between the two electrodes, and the large volumetric change (~80%) between sulfur and $Li_2S$[2–6]. To address these problems, great efforts have been devoted to improve the sulfur utilization and reduce the dissolution of polysulfide intermediates. Many classical strategies, such as sulfur host materials[7–11], protective coating layers[12–16], and interlayer between cathode and separator[17, 18], promoted the springing up of the sulfur cathodes with high-specific capacities. However, it is noteworthy that most of the reported Li–S batteries were constructed with the low-sulfur content ($< 70$ wt%) and/or low-sulfur loading ($< 2$ mg cm$^{-2}$), and the corresponding areal capacities (usually $< 2$ mAh cm$^{-2}$) are smaller than the commercial LIBs ($\sim 4$ mAh cm$^{-2}$)[19–21]. To be comparable with LIBs, a key point for the development of Li–S batteries is how to increase the energy densities of sulfur cathodes while keeping high-sulfur utilization and cycling stabilities. Unfortunately, simply by increasing the sulfur content and sulfur loading, the performance of sulfur cathodes would decline significantly due to the serious downgrade of sulfur utilization[6].

As the lightweight and conductive sulfur hosts, carbon nanomaterials have emerged as the leading candidates for high-sulfur loading. It is realized that large surface area and porous structure can improve the sulfur content in carbon/sulfur composites and physical confinement of the polysulfide intermediates within the carbon frameworks, resulting in high-sulfur utilization[22, 23]. Moreover, to further suppress the dissolution of polysulfide intermediates, heteroatom doping and surface functionalization have been developed as the effective strategies for the promotion of the chemisorption between polysulfide intermediates and carbon nanomaterials[24–27]. However, most of carbon/sulfur cathodes were assembled by the traditional slurry coating process, therefore, the use of conductive agents and binders inevitably offset the energy density advantage of sulfur cathodes even although the sulfur content in many carbon/sulfur composites can easily get over 70 wt%[28]. Recently, a "self-supporting cathode" concept that the free-standing carbon/sulfur composite films can be directly used as the cathodes without using the additives and metallic current collector has been explored to achieve high-sulfur loading[28–44]. Due to the unique self-assembly behaviors of the low-dimensional carbon nanomaterials, the self-supporting cathodes derived from carbon nanofiber[28–31], carbon nanotube[32–35], and graphene[36–39], have upgraded the areal capacities of carbon/sulfur cathodes to a higher level. Unfortunately, the carbon frameworks in these self-supporting cathodes usually exhibited a loose structure with low-specific surface area, thus easily leading to the performance degradation, especially in rate capability and cycling stability[28–44]. These self-supporting cathodes still have to face the common problems caused by increasing the electrode thickness, such as the diffusion pathways, mass transport, and kinetics[6]. More importantly, the imbalance between the areal sulfur loading and electrode thickness makes

the reported self-supporting cathodes high-areal capacities accompanied with low volumetric energy densities, which cannot meet the demand for high-energy-density Li–S batteries[29, 30, 33]. Therefore, the high-sulfur-loading cathode with high-areal and volumetric capacities, long cycle life, and good rate capability, is still a big challenge for Li–S batteries.

Herein, we design a novel two-dimensional (2D) carbon yolk-shell nanostructure, graphene encapsulated in hollow mesoporous carbon nanosheet (G@HMCN), as a promising sulfur host to construct the self-supporting cathodes for Li–S batteries. As a class of fascinating sulfur host materials, hollow carbon nanomaterials (HCN) have been widely used for the fabrication of carbon/sulfur cathodes[45–51]. However, the corresponding HCN/S cathodes are also suffering from the bottleneck problems, including low-sulfur loading, low volumetric energy densities, and poor processing performance[21]. Inspired by the old-fashioned photo album that the printed photographs (i.e., 2D cores) were placed in the plastic pockets (i.e., 2D hollow shells) on a one-to-one basis and then formed a pamphlet with a closely packed structure, we put forward a new concept of 2D carbon yolk-shell nanostructure to overcome the inherent defections of HCN by coupling the advantages of 2D ultrathin nanostructure and hollow porous nanostructure (Fig. 1a). Benefiting from high surface area, large pore volume, and excellent dispersibility, G@HMCN is able to generate 2D carbon/sulfur composite nanosheets (G@HMCN/S) with sulfur content of 80.5 wt%. Simply by vacuum filtration, the co-assembly of G@HMCN/S and graphene can further form a free-standing, flexible, and closely packed G@HMCN/S-G hybrid paper with high-sulfur content (73 wt%) and high-sulfur loading (5 ~ 10 mg cm$^{-2}$). When directly used as the cathodes, G@HMCN/S-G exhibits significantly enhanced performance for high-energy-density Li–S batteries.

## Results

**Synthesis and characterizations of G@HMCN.** Figure 1b illustrated the synthetic procedure for the G@HMCN yolk-shell nanosheets based on a facile hard-templating method. Briefly, graphene oxide (GO) prepared by Hummer method was first coated with a $SiO_2$ layer to yield the GO@$SiO_2$ core-shell nanosheets. And then, the co-assembly of $SiO_2$ and the low-cost carbon precursor polybenzoxazine (PB) onto the surface of GO@$SiO_2$ further formed the sandwich-like GO@$SiO_2$@PB/$SiO_2$ core-shell nanosheets according to our recently developed silica-assisted PB coating strategy[21]. After carbonization under $N_2$ atmosphere and removal of $SiO_2$, GO@$SiO_2$@PB/$SiO_2$ was converted into the highly dispersible G@HMCN. It is well-known that the hard-templating method based on the sol-gel coating technique is a reliable and scalable way to synthesize yolk-shell nanostructures[52, 53]. The intermediates generated in the synthesis process, such as GO@$SiO_2$ and GO@$SiO_2$@PB/$SiO_2$, are highly controllable and reproducible, thus powerfully guaranteeing the synthesis of high-quality G@HMCN (Supplementary Fig. 1).

The low-magnification scanning electron microscopy (SEM) image shows that the as-synthesized G@HMCN products are flat sheets with size range from hundreds of nanometers to several micrometers (Fig. 2a and Supplementary Fig. 2). The high-magnification SEM image further reveals that the surface of G@HMCN is rough and filled with nanopores (Fig. 2b). These densely distributed nanopores were resulted from the removal of $SiO_2$ in the C/$SiO_2$ layer of the carbonization product G@$SiO_2$@C/$SiO_2$, implying that the porous structures of G@HMCN could be controlled during the co-assembly of $SiO_2$ and PB process (see below). Similar to many reported hollow/yolk-shell structures, the edge projection of G@HMCN observed

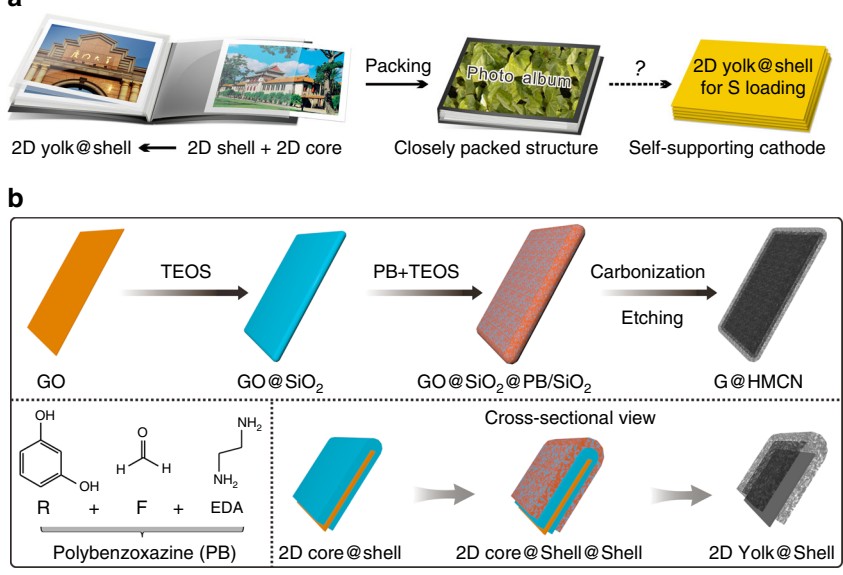

**Fig. 1** Schematic illustration for the synthesis of G@HMCN. **a** Photo album inspired 2D yolk-shell nanostructure for the self-supporting carbon/sulfur cathode. **b** Two-step coating of GO with SiO₂ and porous carbon precursor PB/SiO₂, followed by the carbonization and etching of SiO₂ to produce G@HMCN. Tetraethylorthosilicate (TEOS) was employed as the SiO₂ precursor, and resorcinol (R), formaldehyde (F), and ethylenediamine (EDA), were employed as the PB precursors

from the transmission electron microscope (TEM) image is denser than its internal projection due to the low-density cavity (Fig. 2c)[52, 53]. Interestingly, the basic components for the yolk-shell structures of G@HMCN, including core, shell, and void space between core and shell, can be identified based on the image contrast (Fig. 2c) and ultrathin-section TEM image (Supplementary Fig. 3). In addition, the high-magnification TEM image also clearly indicated the highly porous structure of G@HMCN (Fig. 2d). Since G@HMCN is highly dispersible in water (Fig. 2a), a free-standing G@HMCN paper was obtained by vacuum filtration of the G@HMCN aqueous dispersion (Fig. 2e). More importantly, the as-prepared G@HMCN paper can be bent into a curved structure, making it a potential candidate for bendable electronic devices (Fig. 2f)[28]. The cross-section SEM images of the G@HMCN paper reveal that the individual G@HMCN are stacked into an oriented layered structure (Fig. 2g, h). The thickness of G@HMCN evaluated from these cross-section SEM images is about 40 nm, agreeing with the atomic force microscope (AFM) observation (Supplementary Fig. 4). Moreover, the stacking structure makes it convenient to directly observe some fracture edges of G@HMCN. The typical high-magnification SEM image clearly showed the inner cavity of G@HMCN (Fig. 2h). As a class of graphene-derived 2D porous carbon nanostructures, the G@MCN core-shell nanosheets, have been recently fabricated by directly coating the mesoporous carbon precursors onto the surface of GO templates[54, 55]. Although G@MCN core-shell nanosheets are believed as the promising electrode materials for electrochemical energy storage, the reported G@MCN core-shell nanosheets are highly crimped and aggregated, and thus difficult to directly construct the free-standing paper-like films[54, 55]. In this study, the G@MCN core-shell nanosheets were synthesized in a contrast experiment by the same silica-assisted PB coating process. As expected, vacuum filtration of the PB-based G@MCN aqueous dispersion finally formed the macroscopic particles (Supplementary Fig. 5).

The nitrogen adsorption and desorption measurement was performed to investigate the porosity of G@HMCN (Fig. 2j, k). As shown in Fig. 2j, a type IV isotherm with a high nitrogen uptake at low relative pressure and a hysteresis loop at high relative pressure were attributed to the highly porous structure of G@HMCN, agreeing well with the SEM and TEM observations. The Brunauer-Emmett-Teller (BET) surface area and total pore volume of G@HMCN were estimated to be 2296 m² g⁻¹ and 2.1 cm³ g⁻¹, respectively. The BET surface area of G@HMCN is significantly superior to that of PB-based G@MCN (1387 m² g⁻¹) and G@MCN core-shell nanosheets reported in the literature (Supplementary Fig. 6)[54, 55]. The corresponding pore size distribution calculated using Horvath-Kawazoe (HK) method and Barrett-Joyner-Halenda (BJH) method exhibits three narrow peaks at about 0.8, 1.5, and 3 nm, respectively, indicating that G@HMCN possesses a micro-/mesoporous nanostructure (Fig. 2k and Supplementary Fig. 7). The microporous surface area and volume are 454 m² g⁻¹ and 0.87 cm³ g⁻¹, respectively, which are both lower than their mesoporous counterparts of 1842 m² g⁻¹ and 1.23 cm³ g⁻¹. These data suggested that the high surface area and pore volume of G@HMCN were largely attributed to the SiO₂-derived mesopores. Therefore, the porosity of G@HMCN should be adjustable by the SiO₂/PB layer in the intermediate GO@SiO₂@PB/SiO₂. Benefiting from the highly controllable coating process, the porosity of G@HMCN is easily tailored by adjusting the synthetic parameters, such as the ratio of template GO to PB precursors and the ratio of pore-forming agent (i.e., tetraethylorthosilicate (TEOS) used for the SiO₂/PB layer) to PB precursors. Increasing the dosages of GO and pore-forming agent within a certain range lead to the increase of the BET surface areas and total pore volumes of G@HMCN (Supplementary Fig. 8). Based on the understanding of sulfur host materials, G@HMCN with the highest BET surface area was naturally selected as the target product in this work. Since PB can serve as a precursor for N-doped carbon nanomaterials, the surface chemical composition of G@HMCN was further investigated by X-ray photoelectron spectroscopy (XPS). The components of carbon, nitrogen, and oxygen were detected in XPS survey spectrum, and the corresponding N content of G@HMCN was calculated to be 4.5 at% (Fig. 2l). As shown in Fig. 2m, the N 1s spectrum further shows three characteristic peaks at 398.6, 400.8, and 403.5 eV, respectively, corresponding to pyridinic N, pyrrolic N, and quaternary N doped in the HMCN framework. In

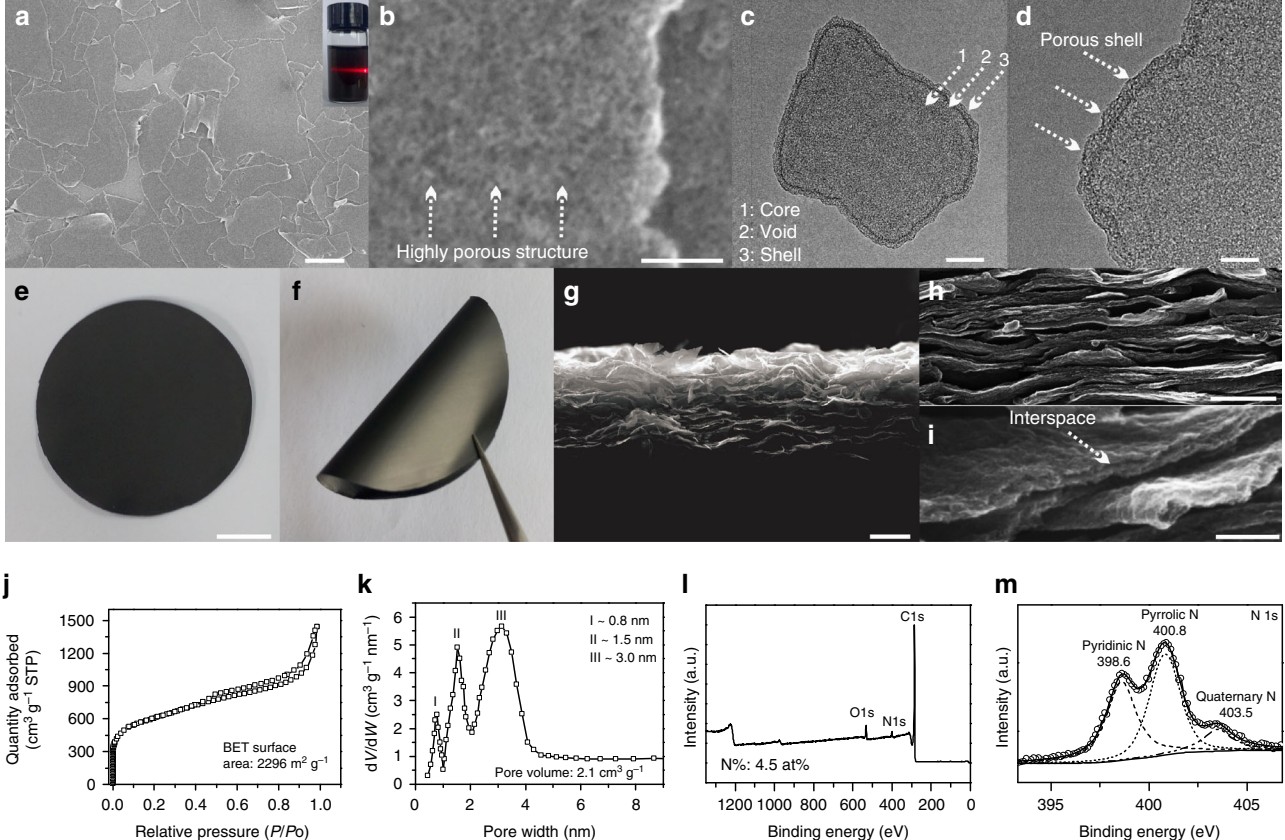

**Fig. 2** Characterizations of G@HMCN. **a, b** SEM images and **c, d** TEM images of G@HMCN; **e, f** Optical images, and **g–i** cross-section SEM images of the flexible G@HMCN paper; **j** $N_2$ sorption isotherm, **k** pore-size distribution, **l** XPS spectrum, and **m** N 1s XPS spectrum of G@HMCN Inset of a: optical image of the G@HMCN aqueous dispersion. Scale bars, 2 μm **a**; 50 nm **b**; 100 nm **c**; 50 nm **d**; 1 cm **e**; 5 μm **g**; 500 nm **h**; 100 nm **i**

addition, oxygen species, such as C=O, C–O, and –OH, were also identified in the C 1s and O 1s spectra, suggesting the oxygen functional groups on the surface of G@HMCN (Supplementary Fig. 9). These nitrogen and oxygen species are beneficial to improve polysulfide binding ability of G@HMCN by the chemisorption interaction (Supplementary Fig. 10)[24–27]. From the viewpoint of carbon/sulfur cathodes, the large surface area, high N content, and surface functional group, implied that G@HMCN is an ideal candidate for high-sulfur loading.

**Construction of the self-supporting carbon/sulfur cathodes.** Until now, reported self-supporting carbon/sulfur cathodes were mostly constructed by impregnating active materials, such as molten sulfur[29], lithium polysulfide solution[36], and sulfur-containing slurry[44], into the prefabricated free-standing carbon films. Unfortunately, when loading high amounts of sulfur, this strategy would inevitably lead the aggregation of sulfur species due to the lack of a controllable way to ensuring the uniform sulfur distribution within the macro-scale carbon frameworks. Since G@HMCN is highly dispersible, providing an opportunity to rationally design the structures of the self-supporting cathodes. The self-supporting cathodes constructed in this work were started with the homogeneous G@HMCN/S composites in order to improve the sulfur loading. As illustrated in Fig. 3a, based on a facile solution-based chemical deposition method, G@HMCN/S composites were synthesized by reacting sodium thiosulfate with hydrochloric acid (i.e., $Na_2S_2O_3 + 2HCl \rightarrow 2NaCl + SO_2 + H_2O + S$) in the G@HMCN aqueous dispersion[12]. Considering the 2D characteristic of G@HMCN/S, 2D graphene nanosheets were selected as the conductive units for construction of the conductive

network, which is beneficial to promote the electrochemical performance of G@HMCN/S-based cathodes[6]. Therefore, the free-standing G@HMCN/S-graphene (G@HMCN/S-G) hybrid papers were further obtained by vacuum filtration of the aqueous dispersion containing G@HMCN/S and graphene with a weight ratio of 9:1 (see section Methods and Supplementary Fig. 11 for further detail).

As shown in Fig. 3b, G@HMCN/S well maintained the structural features and good dispersibility of G@HMCN, and no visible sulfur particles can be observed in the low-magnification SEM image. The corresponding-ray spectroscopy (EDX) spectrum and X-ray diffraction pattern reveal that sulfur was successfully loaded into G@HMCN (Fig. 3b and Supplementary Fig. 12). According to the thermogravimetric analysis (TGA) under nitrogen atmosphere, the sulfur content of G@HMCN/S was calculated to be as high as 80.5 wt% (Fig. 3c). It should be pointed out that 80.5 wt% is a quite high content for the carbon/sulfur composites[19–21]. Theoretically, the pore volume (2.1 $cm^3 g^{-1}$) of G@HMCN can accommodate 81.3 wt% of sulfur ($\rho_s = 2.07 g cm^3$) in the G@HMCN/S composite. Compared with G@HMCN, the low BET surface area (10.6 $m^2 g^{-1}$) and pore volume (0.04 $cm^3 g^{-1}$) of G@HMCN/S indicated that sulfur have been deposited in the porous structures of G@HMCN. To investigate the elemental distribution in the G@HMCN/S, the scanning TEM image and corresponding elemental mapping were further measured (Fig. 3d). The elemental mappings of carbon, nitrogen, and sulfur clearly demonstrated that nitrogen was uniformly doped into the carbon frameworks of G@HMCN and sulfur was uniformly distributed in G@HMCN/S. Moreover, after sulfur loading, the thickness of G@HMCN/S evaluated based on

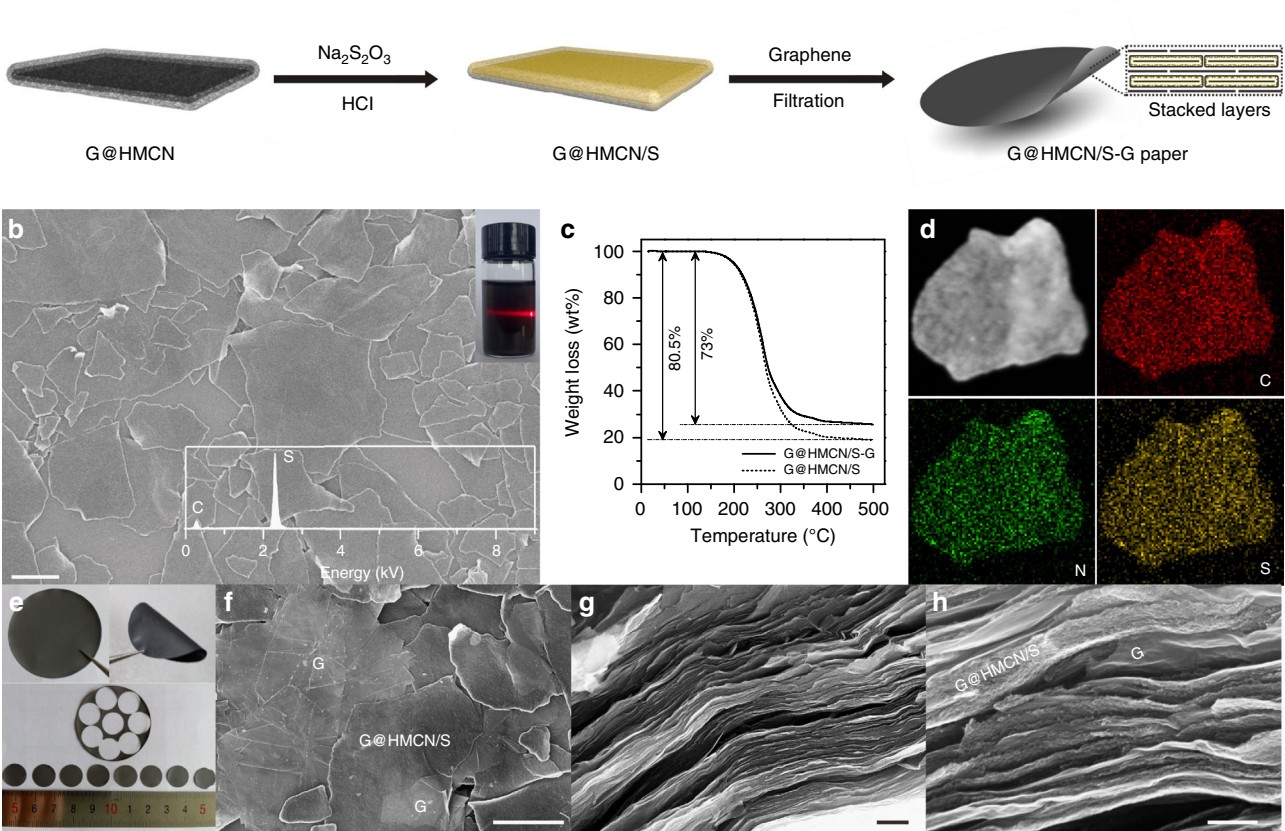

**Fig. 3** Synthesis and characterizations of G@HMCN/S-G. **a** Schematic illustration for the preparation of G@HMCN/S composites and G@HMCN/S-G hybrid paper; **b** SEM image, **c** TGA curve, and **d** scanning TEM image and elemental mapping of G@HMCN/S; **e** Optical images of the G@HMCN/S-G paper and cathodes, **f** top-down and **g**, **h** cross-section SEM images of the G@HMCN/S-G cathodes Insets: the optical image of the G@HMCN/S dispersion and EDX spectrum of G@HMCN/S. Scale bars, 1 μm **b**, **f**, **g**; 200 nm **h**

the AFM image was nearly as thick as G@HMCN and the inner cavity of G@HMCN was still vacant, demonstrating that most of sulfur has been homogeneously distributed in the porous shell (i.e., HMCN layer) of G@HMCN (Supplementary Fig. 4 and 13). Inheriting the merits of G@HMCN and graphene, the free-standing G@HMCN/S-G hybrid paper was highly flexible after completely dried (Fig. 3e). Notably, G@HMCN/S-G hybrid paper can be cut into small pieces, each of which can be directly used as a self-supporting cathode without the metal current collector (Fig. 3e). More importantly, the sulfur content of the G@HMCN/S-G hybrid paper was remained as high as 73 wt% (Fig. 3c), guaranteeing the high-energy-density of the G@HMCN/S-G cathodes[28]. The top-down and cross-section SEM images of the G@HMCN/S-G hybrid paper clearly showed the closely packed G@HMCN/S and graphene with a "face-to-face" mode (Fig. 3f, g and Supplementary Fig. 14). Remarkably, such a brickwall-like structure (illustrated in Fig. 3a) resulted in the high packing density of G@HMCN/S-G ( ~ 1.76 mg cm$^{-3}$). Since packing density and mass transport property is a pair of contradictions, the permeability of G@HMCN/S-G was also verified qualitatively. When used as a filter membrane, the G@HMCN/S-G hybrid paper was permeable to methyl orange molecules, indicating that the penetrable G@HMCN/S-G with high-sulfur content and high packing density is well-suited to the high-energy-density carbon/sulfur cathodes (Supplementary Fig. 15).

**Electrochemical properties**. To evaluate the electrochemical properties, CR2032 coin cells were assembled using the tailored

G@HMCN/S-G as the cathode, and lithium foil as the anode. Since the areal sulfur loading of the G@HMCN/S-G cathode could be easily controlled by adjusting the thickness of the G@HMCN/S-G hybrid paper (i.e., the volume of the mixture of G@HMCN/S and graphene), three typical G@HMCN/S-G cathodes with sulfur loadings of 2.0, 3.5, and 5.0 mg cm$^{-2}$ (designated as G@HMCN/S-G-2.0, −3.5, and −5.0, respectively) were fabricated for comparison. All of the specific capacities calculated in this work were according to the weight of sulfur. Cyclic voltammograms (CV) of the G@HMCN/S-G cathodes were first measured at a scan rate of 0.05 mV s$^{-1}$ (Fig. 4a–c). In the cathodic scans, two peaks (i and ii) at 2.31 and 2.02 V are attributed to the typical multistep reduction process of sulfur from solid $S_8$ to the soluble polysulfides $Li_2S_{4-8}$ to the insoluble products $Li_2S_2/Li_2S^2$. Correspondingly, the two adjacent peaks in the anodic scans (iii and iv) around 2.41 and 2.45 V are derived from the converse oxidation process (i.e., $Li_2S_2/Li_2S$ to $Li_2S_{4-8}$ and then to $S_8$). The slightly widening of the reduction peaks implied that the impedances of the three G@HMCN/S-G cathodes do not increase obviously with an increasing of the sulfur loading[28]. This conclusion was also demonstrated by the electrochemical impedance spectra of the three G@HMCN/S-G cathodes (Supplementary Fig. 16). More importantly, the reduction and oxidation peaks remain almost constant during the three cycles, indicating the good stabilities and reversibilities of the three G@HMCN/S-G cathodes. The galvanostatic charge/discharge behaviors of the G@HMCN/S-G cathodes at current density of 0.2 C (1 C = 1675 mA g$^{-1}$) were further investigated, as shown in Fig. 4d, e, and Supplementary Fig. 17. Two voltage plateaus located around 2.3

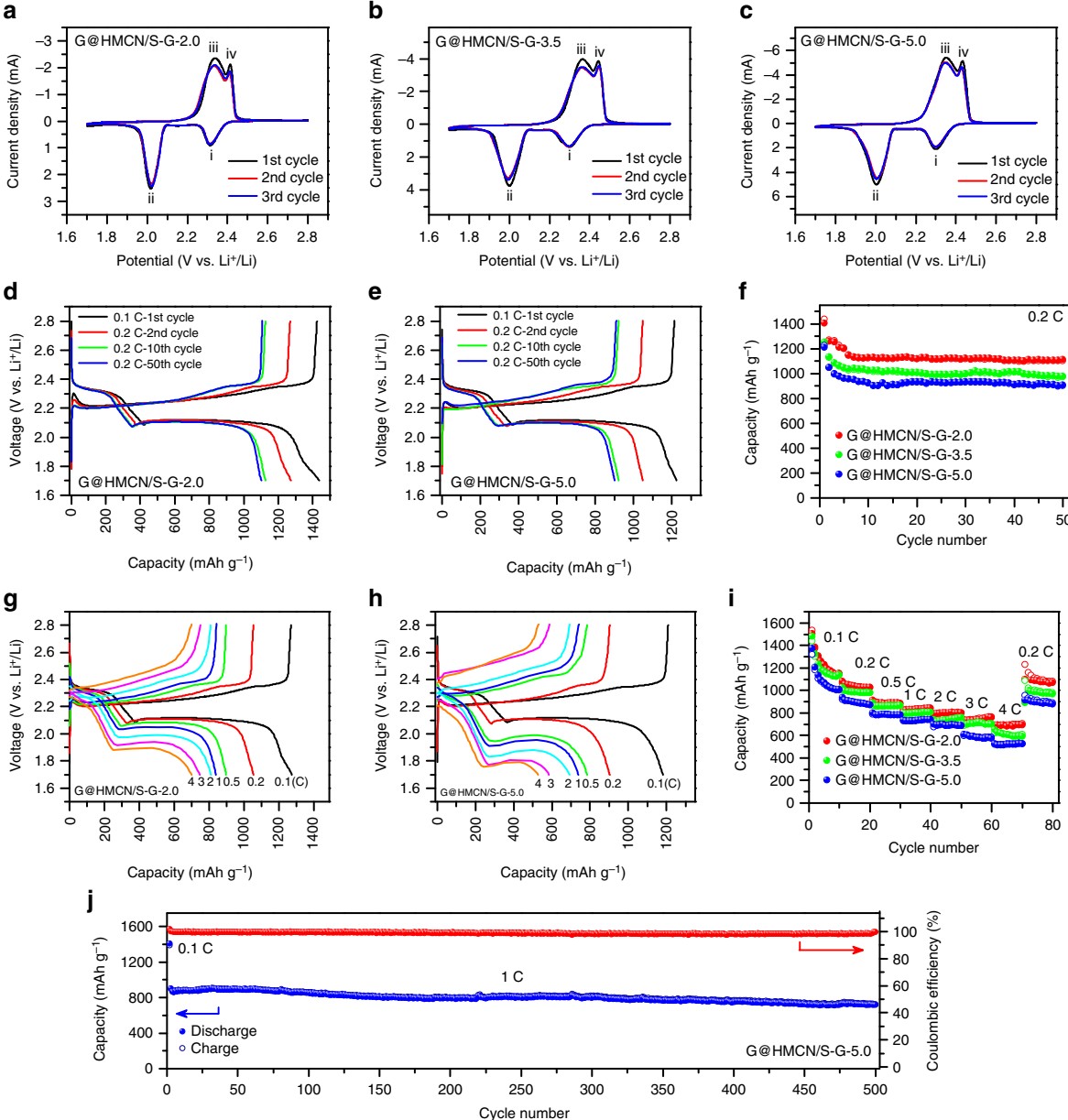

**Fig. 4** Electrochemical properties of the G@HMCN/S-G cathodes. **a–c** CV curves of G@HMCN/S-G-2.0, G@HMCN/S-G-3.5, and G@HMCN/S-G-5.0 cycled at 0.05 mV s⁻¹; **d, e** Charge/discharge curves of G@HMCN/S-G-2.0 and G@HMCN/S-G-5.0 at 0.2 C; **f** Cycling performances of G@HMCN/S-G-2.0, G@HMCN/S-G-3.5, and G@HMCN/S-G-5.0 at 0.2 C; **g, h** Charge/discharge curves of G@HMCN/S-G-2.0 and G@HMCN/S-G-5.0 at various rates; **i** Rate capabilities of G@HMCN/S-G-2.0, G@HMCN/S-G-3.5, and G@HMCN/S-G-5.0; **j** Cycling performance of G@HMCN/S-G-5.0 at 1 C. The solid and hollow symbols in **f, i, j** represent the discharge and charge capacities, respectively

and 2.1 V can be clearly observed in the discharge curves of the G@HMCN/S-G cathodes, agreeing well with the multistep reduction of sulfur revealed by the CV curves[2]. After activation at 0.1 C for one discharge/charge cycle, the initial capacities of G@HMCN/S-G-2.0, G@HMCN/S-G-3.5, and G@HMCN/S-G-5.0 calculated based on the discharge curves at 0.2 C were 1268, 1130, and 1050 mAh g⁻¹, corresponding to the areal capacities of 2.5, 4.0, and 5.3 mAh cm⁻², respectively, indicating the high-sulfur utilization enabled by the frameworks of G@HMCN/S-G. After 50 cycles, the discharge capacities of 1102 (2.2), 972 (3.4), and 903 mAh g⁻¹ (4.5 mAh cm⁻²) were obtained for G@HMCN/S-G-2.0, G@HMCN/S-G-3.5, and G@HMCN/S-G-5.0, respectively, accounting for 87%, 86%, and 86% of the initial capacities, revealing that self-supporting structures designed in this work can effectively alleviate the performance degradation caused by high-

sulfur loading (Fig. 4f)[28]. In addition, the Coulombic efficiencies of the three G@HMCN/S-G cathodes were still more than 99% after 50 cycles at 0.2 C. To further demonstrate that the designed self-supporting structures can effectively entrap the sulfur and polysulfide intermediates at high-sulfur content and high-sulfur loading, the G@HMCN/S-G-5.0 coin cell was further disassembled after 50 cycles. No obvious sulfur agglomerates were observed in the G@HMCN/S-G-5.0 cathode, and the separator exhibited no visible change in color, implying that the sulfur and polysulfides were stably immobilized on the carbon frameworks of the designed self-supporting structures during the lithiation/delithiation process (Supplementary Fig. 18).

To investigate the rate performance, the three G@HMCN/S-G cathodes were further tested by galvanostatic charge/discharge at different current rates, as shown in Fig. 4g, h, and Supplementary

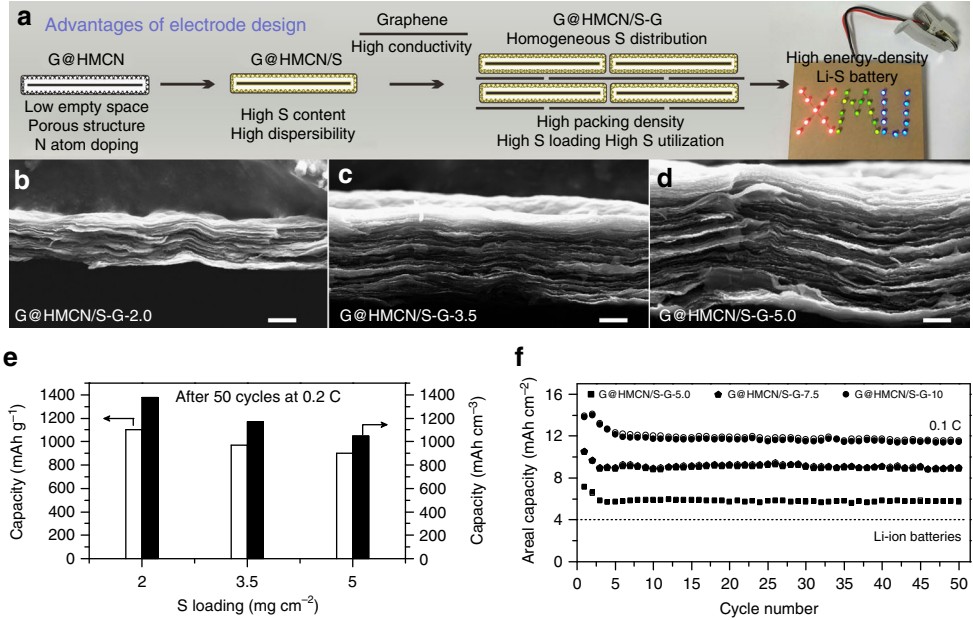

**Fig. 5** The volumetric and areal capacities of G@HMCN/S-G. **a** Schematic illustration for the advantages of electrode design; **b**–**d** the cross-section SEM images of G@HMCN/S-G-2.0, G@HMCN/S-G-3.5, and G@HMCN/S-G-5.0; **e** the volumetric capacities of G@HMCN/S-G-2.0, G@HMCN/S-G-3.5, and G@HMCN/S-G-5.0 after 50 cycles at 0.2 C, **f** the areal capacities of G@HMCN/S-G-5.0, G@HMCN/S-G-7.5, and G@HMCN/S-G-10 at 0.1 C. The solid and hollow symbols in **f** represent the discharge and charge capacities, respectively. Scale bars, 10 μm **b**–**d**

Fig. 19. The charge/discharge profiles from 0.1 to 4 C clearly showed the high-rate performance of the three G@HMCN/S-G cathodes. Take G@HMCN/S-G-5.0 as an example, an obvious discharging platform was still obtained at 1.75 V even at a high rate of 4 C, indicating the facile mass transport and reaction kinetics of the G@HMCN/S-G cathodes. When cycled at 0.1, 0.2. 0.5, 1, 2, 3, and 4 C, the discharge capacities of G@HMCN/S-G-5.0 were 1080 (5.4), 900 (4.5), 786 (3.9), 733 (3.7), 698 (3.5), 593 (3.0), and 524 mAh g$^{-1}$ (2.6 mAh cm$^{-2}$), respectively (Fig. 4i). More importantly, when the current rate is switched back to 0.2 C, G@HMCN/S-G-5.0 delivered a high discharge capacity of 915 mAh g$^{-1}$ (4.6 mAh cm$^{-2}$), demonstrating the excellent stability of G@HMCN/S-G-5.0 at various rates. In addition, the areal capacity of G@HMCN/S-G-5.0 at 4 C was still higher than those of G@HMCN/S-G-3.5 (2.1 mAh cm$^{-2}$) and G@HMCN/S-G-2.0 (1.4 mAh cm$^{-2}$). The polarization of the G@HMCN/S cathode at high current rates increased along with the increasing of the areal sulfur loading, as the overpotentials of cathode and anode were inevitably increased by increasing the areal current densities[28]. Interestingly, the rate performance of G@HMCN/S-G-5.0 was readily improved by shortening the activation process at low current rate. For instance, when rate capability test was carried out from 0.2 C to 4 C after activation at 0.1 C for only one cycle (Supplementary Fig. 20), the polarization of G@HMCN/S-G-5.0 was reduced by decreasing the active material loss occurred during the activation process, and the discharge capacity of G@HMCN/S-G-5.0 at 4 C was raised to 600 mAh g$^{-1}$ (3.0 mAh cm$^{-2}$). At the present stage, a high-sulfur-loading cathode with good rate performance is still not easy to achieve[28–44]. Due to low-sulfur utilization, the capacities of the high-sulfur-loading cathodes often declined rapidly at high-current densities. It is important to point out that the rate performance of G@HMCN/S-G is significantly superior to many slurry-based cathodes and self-supporting cathodes with similar sulfur loading, highlighting the advantage of the G@HMCN/S-G cathodes (Supplementary Tables 1 and 2)[28, 29, 33, 35, 36, 48, 56–58].

To further show the stabilities of the G@HMCN/S-G cathodes, the prolonged cycling performance of G@HMCN/S-G-5.0 was

evaluated at a high rate of 1 C. As shown in Fig. 4j and Supplementary Fig. 21, G@HMCN/S-G-5.0 delivered an initial capacity of 900 mAh g$^{-1}$ after activation at 0.1 C for one cycle. After 500 cycles at 1 C, G@HMCN/S-G-5.0 still remained a high capacity of 719 mAh g$^{-1}$. The capacity retention was calculated to be 79.9%, corresponding to an ultralow capacity decay of 0.040% per cycle. During the whole cycling, the Coulombic efficiency was higher than 99%. The remarkable cycling stability of G@HMCN/S-G-5.0 is outstanding compared to many high-sulfur-loading cathodes reported until now (Supplementary Table 3)[33, 36, 48, 58]. To improve the cycling stability, many high-sulfur-loading cathodes often employed the carbon-based films as the interlayer between the cathode and separator, although it reduced the sulfur content of Li–S batteries[29, 34, 36, 40]. Without using any interlayer, the excellent cycling stability of G@HMCN/S-G-5.0 demonstrated that the self-supporting cathodes designed in this work has a superior ability to trap the polysulfide intermediates at high-sulfur loading. It is worth noting that the corrosion of the Li metal anode was still observed in this work after the long-term cycling at high-areal current densities, as mentioned in the previous report[30]. Therefore, we believe that there is still room to achieve a higher performance by coupling the G@HMCN/S-G cathode with the recent progress of Li metal anode[59].

## Discussion

In general, the structural parameter of the carbon hosts has a huge impact on the electrochemical performance of their carbon/ sulfur composite counterparts. Therefore, a series of carefully designed experiments were carried out to understanding the structural advantages of G@HMCN. As shown in Fig. 1a and Supplementary Fig. 22, G@HMCN can be broken down into two parts: the inner G core (i.e., thermally reduced graphene oxide) and the outer HMCN shell. We successfully synthesized these two structural units by using the thermal reduction of GO@SiO$_2$ under N$_2$ atmosphere and the co-assembly of SiO$_2$ and PB onto the surface of the SiO$_2$ nanosheets (i.e., the nanosheets obtained by calcining the GO@SiO$_2$ under air atmosphere), respectively. The XPS spectra of the as-synthesized G and HMCN clearly

reveal that the doped N atoms of G@HMCN mainly exist in the outer HMCN shell (Supplementary Fig. 22). Under the same process conditions for G@HMCN/S-G-2.0, the capacity and cycling stability of the HMCN/S-G-2.0 cathode (976 mA h g$^{-1}$ after 50 cycles at 0.2 C) were significantly better than those of the G/S-G-2.0 cathode (378 mA h g$^{-1}$ after 50 cycles at 0.2 C), indicating that the HMCN shell of G@HMCN played a crucial role in the immobilization of sulfur and polysulfides (Supplementary Fig. 22). To further dissect the HMCN shell, the BET surface area and N content of G@HMCN were, respectively, reduced by the independent control of the pore-forming agent (i.e., TEOS) and N precursor (i.e., ethylenediamine (EDA)) during the synthesis of GO@SiO$_2$@PB/SiO$_2$ (Supplementary Fig. 23). Without using the TEOS, the G@HMCN/S-G-2.0 cathode assembled by the G@HMCN with a low BET surface area (526 cm$^2$ g$^{-1}$) showed a low capacity of 378 mA h g$^{-1}$ after 50 cycles at 0.2 C, demonstrating that the high surface area is a prerequisite to increase the sulfur utilization. Simply by reducing the dosage of EDA, the G@HMCN/S-G-2.0 cathode assembled by the G@HMCN with a low N content (3.5 wt%) showed a capacity of 826 mA h g$^{-1}$ after 50 cycles at 0.2 C, suggesting that high N content is beneficial to alleviating the shuttle of polysulfides.

Compared with the conventional hollow carbon hosts, the remarkable features of G@HMCN are its shrunken cavity and flake-shaped morphology. To intuitively understand these features, hollow mesoporous carbon spheres (HMCS) and hollow mesoporous carbon hemispheres (HMCH) were further synthesized by the same silica-assisted PB coating strategy, excepting the replacement of the GO@SiO$_2$ nanosheets with the colloidal SiO$_2$ spheres (Supplementary Fig. 24)[21]. With the help of the commercial graphene, the corresponding carbon/sulfur composites (i.e., HMCS/S and HMCH/S) were further used to assemble the HMCS/S-G and HMCH/S-G cathodes according to the process for G@HMCN/S-G. Not surprisingly, at a sulfur loading of 2 mg cm$^{-2}$, the thicknesses of HMCS/S-G-2.0 (51 μm) and HMCH/S-G-2.0 (33 μm) are significantly thicker than that of G@HMCN/S-G-2.0 (16 μm) due to the lager cavities and irregular aggregations of the HMCS/S and HMCH/S composites (Supplementary Fig. 24). As expected, the rate capacities of HMCS/S-G-2.0 (169 mA g$^{-1}$ at 4 C) and HMCH/S-G-2.0 (339 mA g$^{-1}$ at 4 C) are poor than G@HMCN/S-G-2.0 (690 mA g$^{-1}$ at 4 C) due to the increase of the impedance and mass transfer distance with the increase of electrode thickness (Supplementary Fig. 24).

As shown in Fig. 5a, the favorable electrochemical performance of G@HMCN/S-G can be attributed to the following advantages of electrode design: (a) 2D yolk-shell structure can minimize the excess empty space while maintaining the structural advantages of hollow structure; (b) the highly porous G@HMCN can accommodate a large amount of sulfur; (c) the N atom doped in G@HMCN can suppress the shuttling of the soluble polysulfides, realizing the long cycle stability; (d) The high conductivity of graphene in G@HMCN/S-G is beneficial to improve overall electrical conductivity of the self-supporting cathodes; (e) the compact packing of G@HMCN/S and graphene can reduce the mass transfer distance, resulting in the high-rate capability. More importantly, the closely packed structure of G@HMCN/S-G can ensure the high-volumetric capacities. As shown in Fig. 5b–d, the thickness of the three G@HMCN/S-G cathodes observed from the cross-section of SEM images were about 16, 29, and 43 μm, respectively. The corresponding ratio of area sulfur loading to electrode thickness is significantly higher than most of self-supporting cathodes (Supplementary Table 4)[28, 29, 33, 34, 48]. After 50 cycles at 0.2 C, the volumetric capacities of G@HMCN/S-G-2.0, G@HMCN/S-G-3.5, and G@HMCN/S-G-5.0 were calculated to be 1377, 1173, and 1050 mAh cm$^{-3}$, respectively (Fig. 5e). In addition, we also demonstrated that G@HMCN/S-G is capable of

supporting even higher sulfur loading, which is important to meet the requirement for practical application. As shown in in Fig. 5f, the sulfur loadings of the G@HMCN/S-G cathodes were further increased from 5.0 to 7.5 to 10 mg cm$^{-2}$ by controlling the thickness of the G@HMCN/S-G hybrid paper. After 50 cycles at 0.1 C, the discharge capacities of 5.7 (1330), 8.9 (1364) and 11.4 mAh cm$^{-2}$ (1329 mAh cm$^{-3}$) were maintained for G@HMCN/S-G-5.0, G@HMCN/S-G-7.5, and G@HMCN/S-G-10, respectively, indicating the high-areal and volumetric capacities as well as good cycling performance (Supplementary Fig. 25 and Supplementary Table 5). Notably, G@HMCN/S-G provided a favorable tradeoff between the areal and volumetric capacities in comparison to the self-supporting carbon/sulfur cathodes reported until now (Supplementary Fig. 26 and Supplementary Table 6)[29, 30, 33, 35]. Therefore, taking into account of the sulfur content, areal and volumetric capacities, rate capability, and cycle stability, G@HMCN/S-G is a highly promising cathode for advanced high-energy-density Li–S batteries.

In summary, 2D carbon yolk-shell nanomaterial G@HMCN has been successfully designed and synthesized by a facile and reliable hard-templating method. With high surface area, nitrogen doping, unique 2D structure, and high dispersibility, G@HMCN is an ideal platform for the design of high-performance electrode materials. Based on the structural characteristics of G@HMCN, we rationally designed a free-standing and flexible G@HMCN/S-G hybrid paper with high-sulfur loading and sulfur content by the co-assembling of the G@HMCN/S composite and graphene. Since the N-doped G@HMCN, conductive graphene, and closely packed structure of G@HMCN/S-G can greatly improve the sulfur utilization at high-sulfur loading, G@HMCN/S-G with sulfur loading of 5 mg cm$^{-2}$ exhibited a high capacity, good rate performance, and excellent cycling stability accompanied with the favorable balance between the areal and volumetric capacities. More importantly, the sulfur loading of G@HMCN/S-G could be further increased to 10 mg cm$^{-2}$, resulting in a higher areal capacity. We believed that G@HMCN designed in this work not only can provide insights on the high performance Li–S batteries, but also would open opportunities to generate a class of 2D carbon nanomaterials for numerous applications, such as supercapacitors, catalysis and electrocatalysis, and flexible energy storage devices.

## Methods

**Materials**. TEOS, EDA, resorcinol, polyvinyl pyrrolidone (PVP), and Triton X-100 were purchased from Alfa Aesar; ethanol, ammonia aqueous solution (28%), formaldehyde, and sodium thiosulfate were supplied by Sinopharm Chemical Reagent Co.; commercial graphene powder was supplied by Deyang Carbonene Technology Co., Ltd.. All reagents were used without purification. GO was synthesized according to the modified Hummers method[60].

**Synthesis of GO@SiO$_2$ core-shell nanosheets**. Thirty milligram of GO and 0.3 g of PVP were firstly added to a mixture of 120 ml of ethanol and 15 ml of deionized water. After magnetic stirring for 10 min, 6 ml of ammonia aqueous solution and 4 ml of TEOS were added and kept under stirring for 6 h at 30 °C. Then, the resulting dispersion was centrifuged and washed with deionized water. Finally, the as-prepared GO@SiO$_2$ was dispersed into 30 ml of deionized water.

**Synthesis of G@HMCN yolk-shell nanosheets**. Ten milliliters of GO@SiO$_2$ aqueous dispersion was added to a mixture of 60 ml of deionized water and 30 ml of ethanol. After the addition of 0.3 ml of formaldehyde, 0.2 g of resorcinol, 0.3 ml of EDA, and 0.6 ml of TEOS, the mixed dispersion was further stirring at 40 °C for 24 h. The intermediate GO@SiO$_2$@PB/SiO$_2$ was collected by centrifugation and dried at 60 °C for 12 h. After carbonization under N$_2$ atmosphere at 800 °C for 4 h and followed by washing with 10% HF aqueous solution for 24 h, GO@SiO$_2$@PB/SiO$_2$ was finally converted into G@HMCN.

**Synthesis of G@HMCN/S composite**. In all, 0.2 g of G@HMCN was dispersed into 400 ml of deionized water by ultrasonication for 10 mins, followed by the addition of 30 mg of PVP and 7.6 g of Na$_2$S$_2$O$_3$, and then stirring at 25 °C. Then,

100 ml of hydrochloric acid (1 M) was added dropwise at a rate of 10 ml min$^{-1}$ to the G@HMCN/PVP/Na$_2$S$_2$O$_3$ mixture and stirred further for 6 h. The resulting G@HMCN/S composite was collected by centrifugation, washed with deionized water, and finally dried at 60 °C overnight.

**Preparation of the G@HMCN/S-G hybrid paper**. In all, 0.9 g of G@HMCN/S, 0.1 g of commercial graphene powder, and 1 ml of 10 wt% Triton X-100 aqueous solution were dispersed in 500 ml of deionized water by ultrasonication for 0.5 h. Then a certain volume of G@HMCN/S and graphene mixture was filtrated with vacuum pump using a nylon membrane with a pore size of 0.45 μm. After washing with deionized water several times and dried at 60 °C for 5 h, the hybrid paper was pressed at 0.5 MPa and peeled from the filter membrane to get the free-standing G@HMCN/S-G hybrid paper.

**Characterization**. The SEM and TEM images were obtained by Zeiss SIGMA microscope and TECNAI F-30 high-resolution TEM operated at 300 kV, respectively. Nitrogen adsorption and desorption measurement was performed on a TriStar II 3020 system. The specific surface area and pore size distribution curve were calculated by the BET method, HK method, and BJH method, respectively. The XPS spectrum was measured on a PHI QUANTUM 2000. Thermogravimetric analysis was taken using a Pyris Diamond TG-DTA (PE Co., US).

**Electrochemical measurements**. The G@HMCN/S-G hybrid paper was cut and shaped into a circular disc with a diameter of 12 mm and directly used as the cathode without the binder and current collector. CR2032 coin cells were further assembled in an Ar-filled glove box ( < 1 ppm of O$_2$) by using Li foil and Celgard 2400 polypropylene membrane as the anode and separator, respectively. The electrolyte was bis(trifluoromethanesulfonyl)imide lithium (1 M) in a mixed solvent of 1,2-dimethoxyethane and 1,3-dioxolane (v/v = 1:1) with LiNO$_3$ (2 wt%). For better comparison, the ratio of electrolyte to sulfur was controlled as 15 μl mg$^{-1}$ in this work. The electrochemical performance was evaluated using the LAND CT2001A cell test instrument.

**Data availability**. The authors declare that the data supporting the findings of this study are available within the paper.

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

## Acknowledgements

We acknowledge the support from the MOST of China (2015CB932300, 2017YFA0207302), the NSFC (21301144, 21420102001, 21390390, 21333008, 21131005), and the fundamental research funds for the central universities (20720160080).

## Author contributions

X.F. and N.Z. conceived the idea. F.P. and X.F. designed the experiments. F.P. conducted synthesis, characterization, and electrochemical tests. L.L. conducted part of synthesis and electrochemical tests, D.O., Z.Z., and S.M. conducted the TEM and AFM tests. X.F. and N.Z. supervised the project and co-wrote the paper. All authors discussed the results and commented on the manuscript.

## Additional information

**Competing interests:** The authors declare no competing financial interests.

