## [Peer Review File · Nature Communications]

Reviewers' comments:

Reviewer #1 (Remarks to the Author):

This manuscript presents an interesting cathode engineering for building high-performance cathodes for Li-S batteries. It shows high electrochemical performance with high utilization of sulfur and long cycle stability. It also demonstrates excellent engineering performances by showing high-loading sulfur cathode, high-content cathode design, and high volumetric capacities of the corresponding cells. The experimental section provides all the necessary details for repeating the results and for further modification. The manuscript can be published in Nature Communications after addressing the following comments.

1. It is hard to understand which part of the G@HMCN/S cathode plays the key role in improving the cell performances. It is better to clarify the contribution from high surface area, high N content, and functional groups.
2. The porosity analysis shows N₂ absorption behavior resulting from micropores and mesopores, but the range of BJH analysis is too small to tell the microstructure and misses both the mesopore and micropore sections. It is better to identify them and their amount (surface areas and pore volumes).
3. The SEM analysis could not demonstrate the yolk-shell structure. In the EDX, the S signals could result from the surface coating rather a sulfur-yolk structure. Please provide more experimental evidence or analysis to prove it.
4. As the rate performance goes to 1C – 4C, the polarization is too large to imagine that the cell could cycle at such high rates up to 4C and could cycle for 200 cycles at 1C rate. Moreover, although Fig. 4e and Fig. 4g show the performance of G@HMCN.S-G-5.0, their 1C capacities are inconsistent. The

polarization of G@HMCN.S-G-5.0 at 1C rate shown in Fig. 4g and Fig. S 15 are also inconsistent. Why?

5. Most of the cells shown in the comparison table have the sulfur loading less than 10 mg cm⁻². The author might need to reconsider their comparison with the following references: J. Power Sources, 2013, 224, 260; J. Power Sources, 2014, 261, 264; ChenSusChem. 2015, 8, 2892; Energy Environ. Sci., 2016, 9, 3188; ACS Nano, 2016, 10, 10462.

Reviewer #2 (Remarks to the Author):

This work reported a 2D carbon yolk-shell nanosheet (G@HMCN) for high-performance Li-S battery. The G@HMCN/S cathode delivers an excellent rate performance and cycling stability. It is an interesting work and represents a novel prepared process for carbon nanosheet. However, considering vast recent reports involving various carbon structures supported S in Li-S battery field, the idea of this paper is not completely new, so I feel it may be unsuitable for publishing in nature communications.

The more detailed reasons are as follows:

1. Authors claimed that the 2D carbon sheet structures have many advantages to apply in Li-S battery. I feel this view seems to be too subjective, and lacks of convincing evidences after reading through this article. Authors should provide many theoretical and experimental proofs to support the view.

2. Authors think that the loose structure with low specific surface area may easily lead to the degradation for rate capability and cycling stability after combining reference 28-44. This may be a lopsided view and authors should provide some convincing data.

3. In Discussion Section, authors list many advantages of electrode design from (a) to (g). However, among these views, some seem to be repetitive and even very subjective. Authors should refine them. More importantly, more convincing proofs to support these views should be appeared in the Discussion Section.

4. In the research field involving Li-S batteries, many important characterization analyses, such as CV and EIS, can provide more useful information, authors should add these tests to obtain some convincing data.

Reviewer #3 (Remarks to the Author):

In this manuscript, the authors designed a two-dimensional carbon yolk-shell nanosheet to construct a novel self-supporting sulfur cathode, which not only delivers the good rate performance and cycling stability, but also provides a favorable balance between the areal and volumetric capacities. It can be viewed as some progress for superior-capacity freestanding cathodes for high-performance lithium-sulfur batteries, and also contain interesting results. However, there are still several unclear points in the manuscript, and a number of issues should be clarified and certain statements require further justification before it can be accepted. There are given as below:

1. What does G@NHPC/S-G mean here? It is hardly to find where this abbreviation comes from.

2. The authors designed a two-dimensional carbon yolk-shell nanosheet to construct a novel self-supporting sulfur cathode. A self-supporting carbon yolk-shell nanosheet is directly used as a cathode without using the additives and metallic current collector. For the two-dimensional carbon yolk-shell nanosheet, can the author explain the functions of GO and HMCN?

Furthermore, there are already many hybrid carbon structure design for Li-S battery cathode in previous works, so what is the advantage of this hybrid structure comparing with others?

3. In this manuscript, GO (graphene oxides) instead of graphene was used to prepare two-dimensional carbon yolk-shell nanosheet for a sulfur host to construct the self-supporting cathodes. But it is usually believed that the conductivity of GO is poor, so it may not good for facilitating efficient charge transportation as well as electrolyte penetration. How this problem is solved in this work?

4. The nitrogen and oxygen species are beneficial to improve polysulfide binding ability of G@HMCN by the chemisorption interaction, so the authors use PB as a precursor for N-doped carbon nanomaterials. Can the authors explain that where the nitrogen and oxygen species are doped, in GO or HMCN? After all, this is very important to clarify the effects by different carbon materials.

5. The authors also consider the problem of good rate performance for high-sulfur-loading cathode. Both rate performance and cycling performance are not as good as that in many cathodes with lower sulfur loading, which was not listed in for comparison. So does it indicate that much sulfur materials has not been efficiently used in such hybrid structure? Furthermore, as a key point in the manuscript, the amount of sulfur loses during cycling was rarely been mentioned, and it is hard to judge whether the sulfur can stably stay on the carbon materials. Therefore, the authors should pay much more attention on this point.

Reply to Reviewer #1

We would like to thank the referee for the publication recommendation. According to the suggestions given by the referee, we have carefully revised our manuscript:

1. **Comment:** It is hard to understand which part of the G@HMCN/S cathode plays the key role in improving the cell performances. It is better to clarify the contribution from high surface area, high N content, and functional groups.

Response: To understand the contribution from the surface area and high N content, two reference samples G@HMCN with a low BET surface area ($526 \text{ cm}^2 \text{ g}^{-1}$) and G@HMCN with a low N content (3.5 wt%) were synthesized for comparison. Their corresponding electrochemical performances reveal that the high surface area is a prerequisite to increase the sulfur utilization, and high N content is beneficial for the polysulfide immobilization (see Line 9-20 of Page 13 and Supplementary Fig. 22). Theoretically, the oxygen species resulted from the carbonization of oxygen-containing precursors have certain abilities of polysulfide-binding. However, a high content of the residual oxygen species means that the carbonization degree of the precursors is relatively low, which would harm the improvement of the conductivity and surface areas of the carbon nanomaterials. The contents of oxygen species were therefore usually associated with the control of the surface areas and N contents of carbon nanomaterials. In this work, oxygen species provides little guidance for improving the electrochemical properties of G@HMCN compared with high surface area and high N content.

2. **Comment:** The porosity analysis shows N_2 absorption behavior resulting from micropores and mesopores, but the range of BJH analysis is too small to tell the microstructure and misses both the mesopore and micropore sections. It is better to identify them and their amount (surface areas and pore volumes).

Response: According to the reviewers' suggestion, we have re-analyzed the porosity of G@HMCN covering both mesopore and micropore sections (see Line 25-31 of Page 6, Line 1 of Page 7, Fig. 2j, 2K, and Supplementary Fig. 7). Both surface areas and pore volumes contributed from micropores and mesopores are provided in the revised manuscript.

3. **Comment:** The SEM analysis could not demonstrate the yolk-shell structure. In the EDX, the S signals could result from the surface coating rather a sulfur-yolk structure. Please provide more experimental evidence or analysis to prove it.

Response: In the original manuscript, the yolk-shell structure of G@HMCN was identified by the image contrast in TEM image (Fig. 2c). To further demonstrate the yolk-shell structure, the cross-section TEM image of G@HMCN has been provided in the revised manuscript (see Line 21 of Page 5 and Supplementary Fig. 3). Compared with the high surface area and high pore volume of G@HMCN, the decreased surface area and pore volume of G@HMCN/S indicated that most of sulfur was loaded in the porous carbon shell of G@HMCN (see Line 25-30 of Page 8).

4. **Comment:** As the rate performance goes to 1C – 4C, the polarization is too large to imagine that the cell could cycle at such high rates up to 4C and could cycle for 200 cycles at 1C rate. Moreover, although Fig. 4e and Fig. 4g show the performance of G@HMCN/S-G-5.0, their 1C capacities are inconsistent. The polarization of G@HMCN/S-G-5.0 at 1C rate shown in Fig. 4g and Fig. S 15 are also inconsistent. Why?

Response: The large polarization at the high rates is a common phenomenon in the current high-sulfur-loading cathodes. It should be pointed out that the polarization of G@HMCN/S-G-5.0 in the long-term cycling test is superior to many high-sulfur-loading cathodes reported previously (e.g.; *Adv. Energy Mater.* 2015, **5**, 1402263.; *Adv. Mater.* 2015, **27**, 2891.; *Angew. Chem. Int. Ed.* 2015, **54**, 12886.; *Angew. Chem. Int. Ed.* 2016, **55**, 3982.). We also found that the large polarization of G@HMCN/S-G-5.0 in the rate capability test was related to the long-time activation process at 0.1 C. As shown in the following Figure R1, when rate capability test was carried out from 0.2 C to 4 C after activation at 0.1 C for only one cycle (the same activation process for the long-term cycling test at 1 C), the voltage plateaus and capacity of G@HMCN/S-G-5.0 obtained at 1 C rate agrees well with those of G@HMCN/S-G-5.0 obtained from the long-term cycling test.

Figure R1. (a, b) The rate capability test provided in the manuscript, (c, d) the rate capability test carried out from 0.2 C to 4 C after the activation at 0.1 C for only one cycle.

5. Comment: Most of the cells shown in the comparison table have the sulfur loading less than 10 mg cm⁻². The author might need to reconsider their comparison with the following references: J. Power Sources, 2013, 224, 260; J. Power Sources, 2014, 261, 264; ChenSusChem. 2015, 8, 2892; Energy Environ. Sci., 2016, 9, 3188; ACS Nano, 2016, 10, 10462.

Response: According to the reviewers' suggestion, we have added a comparison table for the high-sulfur-loading cathodes (>10 mg cm⁻²) into the revised manuscript (see Supplementary table 5). Considering the sulfur content and cathode area, G@HMCN/S-G-10 developed in this work has an advantage in specific capacity (i.e., sulfur utilization).

Reply to Reviewer #2

We appreciate the constructive comments from the reviewer. Following the comments, we have now carefully revised our manuscript:

1. **Comment:** Authors claimed that the 2D carbon sheet structures have many advantages to apply in Li-S battery. I feel this view seems to be too subjective, and lacks of convincing evidences after reading through this article. Authors should provide many theoretical and experimental proofs to support the view.

Response: To demonstrate the structural advantages of G@HMCN, more control experiments have been designed and provided in the revised manuscript (see Line 9-31 of Page 13, Line 1-7 of Page 14, and Supplementary Fig. 22 and 23). These elaborated experiments cover the major parameters of G@HMCN, such as surface area, N content, cavity size, and geometrical morphology. Due to its ultra-high surface area and high N content, G@HMCN is endowed with the obvious advantages in the sulfur loading and utilization. More importantly, compared with the conventional hollow structures, the shrunken cavity and 2D characteristic of G@HMCN allowed the assembly of G@HMCN/S to possess a high space utilization and high packing efficiency, respectively, resulting in a significant decrease of the electrode thickness and impedance. Therefore, G@HMCN not only conforms to the requirement of the high-performance carbon host materials demonstrated in the literature, but also opens new opportunities to design high-performance self-supporting cathodes with high sulfur content and high sulfur loading.

2. **Comment:** Authors think that the loose structure with low specific surface area may easily lead to the degradation for rate capability and cycling stability after combining reference 28-44. This may be a lopsided view and authors should provide some convincing data.

Response: Since the interface between the carbon host materials and sulfur is the foundation of the sulfur loading and polysulfide immobilization, the high specific surface area of carbon host materials can lead to the high sulfur utilization of the carbon/sulfur composite cathodes. Some self-supporting cathodes thus employed CO₂ as the activation reagent to further increase the surface areas of the carbon matrixes in the self-supporting structures (e.g., *Adv. Mater.* 2015, **27**, 1694.; *ACS Nano* 2016, **10**, 1300.). More importantly, the electrode thickness is another noticeable problem that would influence the mass transport pathways, kinetics and volumetric energy densities of the

high-sulfur-loading cathodes. Due to the increase of mass transfer distance and the decrease of the polysulfide-trapping interface, the high-sulfur-loading cathodes assembled by the loose carbon matrixes with low specific surface area are often difficult to achieve high sulfur utilization with less “shuttle effect”, thus easily resulting in unsatisfactory rate and cycling properties. Based on this point, we designed 2D carbon yolk-shell nanosheets to obtain self-supporting structures with high specific surface area and high packing density. To further demonstrate the impact of the surface area and thickness on the self-supporting cathodes, the reference cathodes were employed and characterized in the revised manuscript (see Line 21-31 of Page 13, Line 1-7 of Page 14, and Supplementary Fig. 23).

3. **Comment:** In Discussion Section, authors list many advantages of electrode design from (a) to (g). However, among these views, some seem to be repetitive and even very subjective. Authors should refine them. More importantly, more convincing proofs to support these views should be appeared in the Discussion Section.

Response: According to the reviewers' suggestion, the descriptions of the advantages of electrode design have been streamlined and optimized in the revised manuscript (see Line 14-16 of Page 14). To support these views, a series of control experiments have been provided in the revised Discussion Section (Page 12-14, and Supplementary Fig. 21, Fig. 22, and Fig. 23).

4. **Comment:** In the research field involving Li-S batteries, many important characterization analyses, such as CV and EIS, can provide more useful information, authors should add these tests to obtain some convincing data.

Response: According to the reviewer' suggestion, the results of the CV and EIS tests have been analyzed and discussed in the revised manuscript (see Line 3-17 of Page 10, Fig.4, and Supplementary Fig. 16), indicating that self-supporting structures designed in this work can alleviate the performance degradation caused by the increase of the sulfur loading.

Reply to Reviewer #3

We appreciate the comments from the reviewer for us to improve the

manuscript. We have now carefully revised our manuscript according to the comments:

1. **Comment:** What does G@NHPC/S-G mean here? It is hardly to find where this abbreviation comes from.

Response: G@NHPC/S-G is a clerical error. In the revised manuscript, G@NHPC/S-G has been replaced with G@HMCN/S-G (see Page 11, 12, 15, and 17).

2. **Comment:** The authors designed a two-dimensional carbon yolk-shell nanosheet to construct a novel self-supporting sulfur cathode. A self-supporting carbon yolk-shell nanosheet is directly used as a cathode without using the additives and metallic current collector. For the two-dimensional carbon yolk-shell nanosheet, can the author explain the functions of GO and HMCN? Furthermore, there are already many hybrid carbon structure design for Li-S battery cathode in previous works, so what is the advantage of this hybrid structure comparing with others?

Response: In this work, 2D ultrathin GO was used as a template for the synthesis of 2D yolk-shell structured G@HMCN. The highly porous HMCN shell of G@HMCN plays a crucial role in the sulfur loading and polysulfides immobilization. The functions of the core and shell of G@HMCN have been discussed in the revised manuscript (see Line 26-31 of Page 12, Line 1-8 of Page 13, and Supplementary Fig. 21). As an upgraded structure of hollow carbon structures, 2D carbon yolk-shell structures can minimize the extra interparticle packing void space while maintaining the structural advantages of hollow structure. Unlike many 2D carbon core-shell structures, 2D carbon yolk-shell structure does not generate agglomeration during the assembly process. By the coupling of 2D ultrathin nanostructure and hollow porous nanostructure, G@HMCN integrated many important parameters, such as ultra-high surface area, large pore volume, and high packing density, providing a new and effective route to fabricate high-performance self-supporting cathodes with high sulfur content and high sulfur loading. Based on the structural advantages of G@HMCN, the G@HMCN/S-G cathodes exhibited a superior balance between the areal and volumetric capacities in comparison to many hybrid carbon structures designed in the previous works.

3. **Comment:** In this manuscript, GO (graphene oxides) instead of graphene was used to prepare two-dimensional carbon yolk-shell nanosheet for a sulfur host to construct the self-supporting cathodes. But it is usually believed that the conductivity of GO is poor, so it may not good for facilitating efficient charge transportation as well as electrolyte penetration. How this problem is solved in this work?

Response: In this work, GO in the $\text{GO@SiO}_2\text{/PB/SiO}_2$ has been thermally converted into reduced graphene oxides during the carbonization process. The conductivity of thermally reduced graphene oxides is significantly higher than that of GO. The G@HMCN/S-G cathodes are highly penetrable, which has been verified by the vacuum filtration of the methyl orange solution (see Supplementary Fig. 15). Therefore, the electrolyte penetration in the G@HMCN/S-G was realized mainly through the interspace between the G@HMCN/S and commercial graphene (see Supplementary Fig. 15).

4. **Comment:** The nitrogen and oxygen species are beneficial to improve polysulfide binding ability of G@HMCN by the chemisorption interaction, so the authors use PB as a precursor for N-doped carbon nanomaterials. Can the authors explain that where the nitrogen and oxygen species are doped, in GO or HMCN? After all, this is very important to clarify the effects by different carbon materials.

Response: The nitrogen species of G@HMCN originate from the carbon precursor polybenzoxazine. Both G and HMCN were subjected to XPS analysis. The results indicated that the nitrogen species of G@HMCN mainly exist in HMCN (see Supplementary Fig. 21). Since the oxygen species of G@HMCN are derived from both GO and polybenzoxazine, the residual oxygen species existed both in the G core and HMCN shell after carbonization process (see Supplementary Fig. 21).

5. **Comment:** The authors also consider the problem of good rate performance for high-sulfur-loading cathode. Both rate performance and cycling performance are not as good as that in many cathodes with lower sulfur loading, which was not listed in for comparison. So does it indicate that much

sulfur materials has not been efficiently used in such hybrid structure? Furthermore, as a key point in the manuscript, the amount of sulfur losses during cycling was rarely been mentioned, and it is hard to judge whether the sulfur can stably stay on the carbon materials. Therefore, the authors should pay much more attention on this point.

Response: Considering the high sulfur content of the cathodes (73%), the capacities and rate capabilities of the G@HMCN/S-G cathodes is far superior to many carbon/sulfur cathodes reported previously, indicating the high sulfur utilization. According to the reviewers' suggestion, the comparison between the G@HMCN/S-2.0 cathode and other carbon/sulfur cathodes with low sulfur loading has been provided in the revised manuscript (see Supplementary Table 2). The high capacity retention and ultralow capacity decay (~0.040% per cycle) of G@HMCN/S-G-5.0 during the 500 cycles at 1 C reveals that the loss of the active materials in G@HMCN/S-G-5.0 is lower than many other high-sulfur-loading cathodes. In the revised manuscript, the disassembly and characterization of the G@HMCN/S-G-5.0 coin cell after the cyclic test at 0.2 C further demonstrated that the sulfur and polysulfides were stably immobilized on the carbon frameworks of the designed self-supporting structures during the lithiation/delithiation process (see Line 3-11 of Page 12 and Supplementary Fig. 18).

Reviewers' comments:

Reviewer #1 (Remarks to the Author):

This manuscript presents interesting cathode engineering for building high-performance cathodes for Li-S batteries. The authors have responded well to the comments in their revision. The authors further provide lots of important finding in support of their analytical results. A few comments are listed below.

1. The authors have responded well to comment 1 of the reviewer 1. Fig. S22 provides important information about what kinds of modification is necessary and what kind of modifications are not needed for developing the Li-S battery technology.
2. The authors have responded to comment 2 of reviewer 1. A minor comment is that the surface areas and pore volumes of various pores (micropores, mesopores, and macropores) are still not provided in the revised manuscript.
3. The authors have responded to comment 4 of reviewer 1. The analysis of the relationship between the rate-performance and polarization is very important and rarely mentioned in Li-S research. It is strongly suggested that the authors provide their findings to the community.

Reviewer #2 (Remarks to the Author):

I suggest this paper can be accepted.

Reviewer #3 (Remarks to the Author):

The authors have revised and improve the manuscript carefully, and I think all the problems the reviewers concerned have been solved. Now it can be published as it is.

Reply to Reviewer #1

1. **Comment:** The authors have responded well to comment 1 of the reviewer 1. Fig. S22 provides important information about what kinds of modification is necessary and what kind of modifications are not needed for developing the Li-S battery technology.

Response: Thank the reviewer for the comment.

2. **Comment:** The authors have responded to comment 2 of reviewer 1. A minor comment is that the surface areas and pore volumes of various pores (micropores, mesopores, and macropores) are still not provided in the revised manuscript.

Response: Both SEM and TEM analyses revealed that the surface of G@HMCN is filled with nanopores whose pore size is far less than 10 nm. The observations agree well with the nitrogen adsorption and desorption measurement. Moreover, the closely packed structure of the G@HMCN paper significantly reduces the void space between the adjacent G@HMCN nanosheets, preventing the formation of macropores. Therefore, the surface areas and pore volumes of G@HMCN mostly originate from the micropores and mesopores. In our last revision, we provided the BET surface area, total pore volume, micropore surface area, and micropore pore volume of G@HMCN. According to the reviewers' suggestion, we now also include both data of mesopore surface area and mesopore pore volume of G@HMCN in the revised main manuscript (see Line 28-31 of Page 6)

3. **Comment:** The authors have responded to comment 4 of reviewer 1. The analysis of the relationship between the rate-performance and polarization is very important and rarely mentioned in Li-S research. It is strongly suggested that the authors provide their findings to the community.

Response: According to the reviewers' suggestion, the additional experiment and the corresponding discussion provided in our response of previous comment 4 are included in the revised manuscript to reveal the relationship

between the rate-performance and polarization of the G@HMCN/S cathode (see Line 29-31 of Page 11, 1-7 of Page 12, and Supplementary Fig. 20).

Reviewers' comments:

Reviewer #1 (Remarks to the Author):

The manuscript can be accepted for publication.

Reviewer #4 (Remarks to the Author):

Lithium-sulfur is an attractive battery chemistry that has been intensively pursued in the past years due to its high theoretical specific energy. Stable battery cycling for sulfur cathodes with both high areal capacity loading and high volumetric capacity is vital to make lithium-sulfur battery a viable technology. However, it is very challenging. In this manuscript, a 2D carbon yolk-shell nanosheet was developed as a lightweight and conductive sulfur host to construct a novel self-supporting sulfur cathode with high sulfur loading and high sulfur content. The batteries showed a favorable balance between the areal and volumetric capacities, and stable battery cycling, which outperform most of the current studies. Meanwhile, the reasons for the good electrochemical performance were discussed. In the respect, this work gets high marks. This is a very well presented and crisp study on high-energy-density lithium-sulfur batteries. This work is of broad enough interest that it will attract the battery community. I recommend this manuscript for the publication in Nature Communication with a small revision according to the following aspects:

1. Line 29, Page 11. The authors should explain the difference in the polarization of the G@HMCN/S cathode with the increasing of the areal sulfur loading clearly. Actually, with the increasing of the areal sulfur loading, the areal current density of both the sulfur cathode and lithium metal anode increases, although the electrodes with different sulfur loading has the same C rate. Large overpotential is produced both at the sulfur cathode and lithium metal anode at high areal current densities.

2. Stable cycling for 500 cycles was achieved for a sulfur cathode with 5 mg cm⁻² sulfur loading at 1 C. Challenges for long cycle life of lithium-sulfur batteries arise from both the cathode and the anode. How is the Li metal anode after many cycles at such a high areal current density and areal capacity?

Reply to Reviewer #4

1. **Comment:** Line 29, Page 11. The authors should explain the difference in the polarization of the G@HMCN/S cathode with the increasing of the areal sulfur loading clearly. Actually, with the increasing of the areal sulfur loading, the areal current density of both the sulfur cathode and lithium metal anode increases, although the electrodes with different sulfur loading has the same C rate. Large overpotential is produced both at the sulfur cathode and lithium metal anode at high areal current densities.

Response: Thank the reviewer for the helpful suggestion. The large overpotential produced at high areal current densities is an important reason for the large polarization of high-sulfur-loading G@HMCN/S cathode, which has been provided in the revised manuscript (see Line 31 of Page 11, 1 of Page 12).

2. **Comment:** Stable cycling for 500 cycles was achieved for a sulfur cathode with 5 mg cm⁻² sulfur loading at 1 C. Challenges for long cycle life of lithium-sulfur batteries arise from both the cathode and the anode. How is the Li metal anode after many cycles at such a high areal current density and areal capacity?

Response: Although the high-sulfur-loading cathode designed in this work has a superior ability to trap the polysulfide intermediates, the corrosion of the Li metal anode was still observed after the long-term cycling test. We fully endorse your view that challenges for long cycle life of lithium-sulfur batteries

arise from both the cathode and the anode. Actually, the high-performance Li metal anode is an important direction for future development of lithium-sulfur batteries. The corresponding discussion has been included in the revised manuscript (see Line 1-5 of Page 13 and Ref. 59).